# Development and Characterization of a New TILLING Population for Forward and Reverse Genetics in Barley (*Hordeum vulgare* L.)

**DOI:** 10.3390/plants13172490

**Published:** 2024-09-05

**Authors:** Feifei Wang, Liang Zhu, Zhenxiang Zhou, Yangyang Gu, Baojian Guo, Chao Lv, Juan Zhu, Xiaohui Liu, Rugen Xu

**Affiliations:** 1Key Laboratory of Plant Functional Genomics of the Ministry of Education, Jiangsu Key Laboratory of Crop Genomics and Molecular Breeding, Jiangsu Co-Innovation Center for Modern Production Technology of Grain Crops, Institutes of Agricultural Science, Yangzhou University, Yangzhou 225009, China; feifei.wang@yzu.edu.cn (F.W.); bjguo@yzu.edu.cn (B.G.); clv@yzu.edu.cn (C.L.); 007670@yzu.edu.cn (J.Z.); 2College of Food and Pharmaceutical Engineering, Guizhou Institute of Technology, Guiyang 550003, China

**Keywords:** barley, mutagenesis, TILLING, mutant phenotype, *HvGLR3.5*

## Abstract

Mutagenesis is an important tool in crop improvement and free of the regulatory restrictions imposed on genetically modified organisms. Barley (*Hordeum vulgare* L.) is a diploid species with a genome smaller than those of other members of the Triticeae crops, making it an attractive model for genetic studies in Triticeae crops. In this study, we report an ethyl methane sulfonate (EMS)-mutagenized population in the Chinese barley landrace TX9425, which is tolerant to both abiotic and biotic stress. A TILLING (Targeting Induced Locus Lesion in Genomes) population consisting of 2000 M_2_ lines was also constructed based on the CEL I enzyme with subsequent polyacrylamide electrophoresis, which decreased the cost and labor investment. The mutant phenotypes of the M_2_ and M_3_ generations were scored and revealed the presence of a wide spectrum of morphological diversity. The population was evaluated by screening for induced mutations in five genes of interest. A detailed analysis was performed for the *HvGLR3.5* gene and three mutations were identified by screening in 2000 M_2_ lines. Two of three mutations displayed tuft and yellow striped leaves compared to the wild type. Altogether, our study shows the efficiency of screening and the great potential of the new TILLING population for genetic studies in the barley crop model system.

## 1. Introduction

The genus *Hordeum* displays a high degree of adaptation to different stressful environments [1], and cultivated barley is grown in subtropical, temperate, and subarctic areas, from sea level to the Andes and the Himalayas [2]. With such a great diffusion as well as abundant genetic stocks with over 485,000 germplasm accessions [3], the barley gene pool contains characters for wide environmental adaptability and strong stress resistance [4].

Mutation is the ultimate source of all genetic variations which provides the raw materials for natural selection and a driving force in evolution. Mutations arising spontaneously are random events and many of the mutations may confer no immediate advantage at the beginning but may help to generate a wide range of useful recombinant genotypes through the subsequent process of independent segregation and crossing over of genes [5]. Historically, the first natural mutant plant in cereals was found in China about 2317 years ago [6]. By now, extensive use of mutation breeding has already made significant contributions to crop production. Over the past 85 years, more than 3000 varieties were derived from the mutagenesis program, with 873 rice lines, 307 barley lines, and 265 wheat lines as the top crops listed in the IAEA/FAO mutant variety database (https://nucleus.iaea.org/sites/mvd/SitePages/Home.aspx, accessed on 18 August 2024). Various effective methods are available to induce mutations, which are categorized as physical or chemical mutagens, such as gamma rays, ethyl methane sulfonate (EMS), and sodium azide (NaN_3_) [7]. With the recent advances in genomics, the use of high-throughput platforms such as TILLING (Targeting Induced Local Lesions in Genomes) in the highly efficient evaluation of mutant pools for specific genomic sequence alterations is crucial in studying the genetic variability at a molecular level [8]. The TILLING method is economical and efficient for screening mutants, which is based on conventional PCR. The subsequent mutation detection is followed by any suitable method including denaturing high-pressure liquid chromatography, fluorescence-based gel electrophoresis (e.g., LI-COR), or unlabeled capillary electrophoresis of heteroduplexes digested with the endonuclease CEL I [9,10].

Up to now, several barley TILLING populations have been developed, including the cultivar Golden Promise population as the reference genotype for barley transformation, with 3072 M_2_ mutants [11]. Malting cultivar Barke was used to generate a TILLING resource comprising 10,279 M_2_ mutants [12], and leaf material and seeds from approximately 20,000 M_2_ plants in cultivar Optic were individually harvested to develop its mutant population [10]. Recently, a landrace “Hatiexi” mutagenic population consisting of 8525 M_3_ lines was complemented with a high-quality de novo assembly of a reference genome for this genotype [13]. These resources combined with a high-quality genome reference sequence have enabled the functional validation of a list of candidate genes resulting from forward genetics and mapping [14,15,16]. In this study, we developed an EMS-induced mutant population in a Chinese barley landrace “TX9425” (TX) and generated a TILLING population with 2000 M_2_ mutants. Here, we evaluated and described various features of our TX mutant population in the M_2_ and M_3_ generations, including its development, mutation genotypes, and a detailed analysis that was performed for the glutamate receptor-like (GLR) gene *HvGLR3.5*. Our results suggested that the new TILLING resource of the barley cultivar TX had potential for use in fundamental research as well as for applied breeding.

## 2. Materials and Methods

### 2.1. Plant Materials

TX9425 (TX), which was one of the Chinese two-rowed landraces, was used to generate the ethyl methane sulfonate (EMS)-induced population. TX was proved to be resistant to salt and waterlogging stress but had a low malting quality. In contrast, a Japanese two-rowed malting barley, NasoNijo (NN), with good agronomic traits and sensitivities to salt and waterlogging stress, was also studied in this experiment. Barley grains of TX and NN were harvested in the field of Yangzhou University, and the grain length, width, and thousand-grain weight were detected by the WSeen SC-G automatic seed selection and analysis system (Model SC-G, Wanshen Ltd., Hangzhou, China). The total nitrogen content of the grain was measured by a semi-automatic Kjeldahl nitrogen analyzer (FOSSKjeltec8100, FOSS, Hillerød, Denmark) according to the Kjeldahl method [17].

### 2.2. Generation of EMS-Mutagenized Population

A total of 0.5 kg of barley seeds (approximately 10,000) was soaked in distilled water for one night and then mutagenized in 0.5% EMS (liquid, Sigma, St. Louis, MO, USA) for 8 h at room temperature. The seeds were rinsed continuously under running water for 3 h to wash off excess EMS and then laid flat on a plate and covered with moist filter paper. Seed germination was recorded as the proportion of germinated seeds per total number of seeds and germinated seeds (M_0_) that were sown in the field in Yangzhou with an interval of 5 cm between seeds. M_1_ seeds were collected from a single plant and grown in the field. The two 4 cm length leaf segments of each individual M_2_ seedling were sampled: one was for genomic DNA extraction with the CTAB method and another one was kept frozen as a backup. The phenotypes of M_2_ seedlings totaled to 29 traits that were recorded during the whole growth period, which include seedling habit (such as albino, etiolated), leaf shape (such as wide, thin, curled), leaf color (such as light leaf color, yellow), tillering (such as non-tiller, multiple tillers, less tiller), plant height (such as dwarf), and spikes (such as six rows, spike anomaly, white glume). The mutation frequency observed in the M_2_ population was calculated by the number of M_2_ mutants/total M_2_ seedlings × 100%. The M_3_ generation was grown for genotyping and the phenotypes recorded in the field and the seeds from the sampled M_2_ seedlings were collected, indexed, and archived. The phenotypes of M_3_ seedlings including plant height, spike length, spacer number, spike number, and thousand-kernel weight were recorded.

### 2.3. Establishment of the CEL I Enzyme Digestion System

The extraction of the CEL I enzyme was based on the method of Till et al. [9]. Then, the activity of the CEL I and enzyme digestion system were verified by the barley gene *HvWRKY62*. In the CEL I enzyme activity experiment, two plasmids (encoding the *HvWRKY62* gene) which contained a single base difference were used as PCR amplification templates, and the PCR products were separated by CEL I enzyme digestion and gel electrophoresis (agarose). In the enzyme digestion system experiment, the extracted enzyme and the DNA of M_2_ samples were diluted to different concentrations to verify the best enzyme digestion system. Genomic DNA of M_2_ individual samples was extracted and pooled with 5 samples.

### 2.4. Screening of the Mutation Lines

The primers of target genes were designed using the NCBI primer-BLAST website (https://www.ncbi.nlm.nih.gov/tools/primer-blast/index.cgi, accessed on 10 January 2024). The PCR reactions were conducted using a thermal cycler (T100 Thermal Cycler, Bio-rad, Hercules, CA, USA) as follows: heat denaturation at 95 °C for 3 min, followed by 35 cycles (94 °C for 30 s, 60 °C for 60 s, 72 °C for 1 min), 72 °C for 5 min, and 95 °C for 5 min; 95 °C–85 °C, −1 °C/15 s; 85 °C–25 °C, −0.1 °C/2 s. The PCR product was subjected to CEL I enzyme digestion at 45 °C for 45 min and then added to the termination solution to terminate the reaction. The CEL I enzyme digestion system included the following: 10 μL of the PCR product, 1 μL of the CEL I enzyme, 2 μL of the CEL I digestion buffer, and 7 μL of ddH_2_O. The enzyme digestion product was then detected by 8% polyacrylamide gel. For confirmation of presumed mutated loci, amplicons of the respective target gene were generated from putative mutants by utilizing the same PCR conditions as established for CEL I analysis and sent to the company for sequencing.

### 2.5. Phenotype Experiment

The seeds of TX, mutant line FM10587, and FM10811 were grown in pots (8 cm × 12 cm) in a greenhouse with a day/night temperature of 22 ± 3 °C in a 16 h/8 h day/night regime. The plant height (from the base of the plant to the tip of the flag leaf), root length (from the base of the plant to the tip of the longest root), and relative chlorophyll content (SPAD value) were measured after four weeks.

## 3. Results

### 3.1. Generation of a TILLING Population in the Barley Cultivar “TX9425”

The two-rowed barley cultivar TX9425 (TX) was used to develop a mutant population. TX is characterized by various traits including salinity tolerance [18,19], waterlogging tolerance [18,20,21], disease resistance [22,23], and dwarf genes [24]. However, this variety also showed some unfavorable traits, including thicker husk, short and twisty spike length (Figure 1A,B), and high grain density [25,26]. Compared with NasoNijo (NN), which has good agronomic traits but is sensitive to salt and waterlogging stress, the thousand-kernel weight of TX is higher than NN (Figure 1C) and the grain of TX is fuller than NN with a shorter grain length and much wider grain width (Figure 1C,E,F). The nitrogen content in the grain of TX was lower than NN (Figure 1G).

Based on previous experience with EMS mutagenesis in a different barley cultivar [10,12], we established a population of mutants in the TX background by treating seeds with 0.5% *v*/*v* EMS for 8 h. The germination rate of M_1_ seedlings was around 70%, and the leaves of some M_1_ seedlings grown in the field were albino and etiolated, which decreased the survival rate of M_1_ seedlings to around 60%. Finally, 1022 M_1_ plants advanced to the M_2_ generation. We sowed all the M_2_ lines and collected leaf samples from a single seedling for DNA extraction and thereby obtained a library with 2000 high-quality M_2_ DNA samples.

### 3.2. Mutant Types and Mutation Frequency Observed in the M_2_ Population

In total, 7224 M_2_ individuals were scored for their phenotypes during the field cultivation by visual assessment of 29 traits for plant seedlings, leaves, stems, and spikes. The results showed that TX produced abundant phenotypic variations after EMS mutagenesis, and the seedlings were divided into mutated individual lines in terms of seedling growth habits, leaf shape, leaf color, tiller number, plant height, spike type, and row type (Figure 2 and Table 1). A total of 319 mutant lines were found in the M_2_ generation, with a mutation frequency of 4.42%. At seedling stage, a total of three types of mutation were found in the seedling habits of 48 strains (Table 1). Among them, there were 12 lines with etiolated leaves in the whole plant or growth period of seedlings with a mutation frequency of 0.17%, and 34 mutant lines with a mutation frequency of 0.47% showed a seedling tuft phenotype (Table 1). In leaf phenotypes, this population in the M_2_ generation displayed abundant phenotypes which included leaf yellow stripes, light leaf color, yellow leaf, striata, flag leaf deformation, leaf spot, and wide, thin, curled and non-midribs (Figure 2 and Table 1). Among these mutation types, the mutation frequency of yellow leaf and leaf yellow stripes were higher than other mutation types with 0.57% and 0.50%, respectively (Table 1). The mutation of yellow leaf stripes occurred at the tillering stage, and the chlorophyll contents at the edges or middle of some leaves were significantly reduced. Another mutation type which is similar to yellow leaf stripes was white stripes with a low mutation frequency of 0.06% (Table 1). In the M_2_ generation, there were 75 lines related to stem mutation type, with a total mutation frequency of 1.03% (Table 1), and the less-tillering phenotype showed the highest mutation frequency compared with other phenotypes (Table 1). Phenotypes related to spike mutations were investigated, which included six rows, spike anomaly, white glume, red glume, long awn, short awn, awnless, and sterile (Figure 2 and Table 1). Among these phenotypes, the mutation frequency of spike anomaly and white glume was relatively higher than other phenotypes (Table 1).

### 3.3. The Main Agronomic Traits of Stable Mutant Lines in the M_3_ Generation

The main agronomic traits of stable mutant lines in the M_3_ generation were investigated under field conditions from the subsequent year, and the results showed that there were abundant mutation phenotypes in plant height, spike length, spike number, and thousand-kernel weight (Table 2). In the M_3_ generation, the plant height was in the range from 27.72 to 91.50 cm; the spike length was in the range from 2.50 to 12.03 cm; the spike number was in the range from 2.78 to 15.56 cm; and the thousand-kernel weight was in the range from 15.63 to 48.39 g (Table 2). Among these agronomic traits, the coefficient of variation (CV) in spike number was the highest with 43.04% (Table 2).

### 3.4. Mutant Screening

Before the mutant screening from the TILLING population, the CEL Ⅰ enzyme digestion system was established. In this experiment, the CEL I enzyme was extracted from fresh celery in Yangzhou, and each time, the enzymatic activity was detected after extraction to ensure that it had a good enzymatic effect during the digestion process. Two plasmids (PW1 and PW2) containing the same *HvWRKY62* gene (with a single base difference) were used as PCR amplification templates, and the PCR products were separated by CEL I digestion and gel electrophoresis. The two template plasmids were amplified to 1327 bp products and digested to produce 371 bp and 956 bp fragments. The amplification primers, differential bases, and fragment sizes are listed in the Appendix A. After confirming the activity of the CEL I enzyme, the effect of the CEL I enzyme concentration on the enzyme digestion was analyzed, and the enzyme digestion system was optimized. In this experiment, the extracted CEL I enzyme was diluted into different concentrations (1×, 0.5×, 0.25×, 0.1×, 0.05×, 0.025×, 0.0001×, H_2_O) to explore the optimal concentration of the CEL I enzyme. Then, the two plasmids were mixed in ratio of 1:1, 1:2, 1:4, and 1:8, respectively, to determine the suitable barley DNA pool. Based on these results (Appendix A), a 1× concentration of the CEL I enzyme was best for the digestion and the number of mixed DNA samples was five.

A total of 400 DNA pools were constructed in groups of five M_2_ DNA samples to screen for mutants. The screen revealed a total of 11 mutant strains from the EMS mutant pool, including *HvRBOHB*, *HvRBOHE*, *HvDTX16*, *HvGLR2.8*, and *HvGLR3.5* genes (Table 3). In these mutant lines, there was one mutation located in the intron; six of the exon-located mutations did not induce a change in the amino acid (AA) sequence, and four induced a change in AAs in the protein sequence (Table 3).

### 3.5. New Alleles of Functional Mutations in the Gene HvGLR3.5

A detailed analysis was performed for the *HvGLR3.5* (HORVU.MOREX.r3.7HG0733700.2) gene that belongs to the glutamate receptor-like (GLR) family of ion channels [27], and GLR3.5 functions as an off switch that blocks the transmission of wound-induced electrical potentials [28]. *HvGLR3.5* had six exons and five introns, and three mutations were found in Exon 2 (Figure 3A). Two of three *Hvglr3.5* mutants (FM10811 and FM10587) showed mutant phenotypes different to wild-type TX (Figure 3C–E). At the one-month seedling stage, phenotype measurements displayed that the relative chlorophyll content (SPAD), root length, and plant height of the FM10587 mutant were all lower than wild-type TX (Figure 3B). Compared with the wild type, these two homozygous mutants showed a significantly reduced plant height, both at the seedling stage and in the field (Figure 3B,F,G). In the field observation, the FM10811 variety showed a tuft phenotype with more tillers and lower plant height compared to TX. Meanwhile, the FM10587 mutant displayed yellow striped leaves (Figure 3F,G).

## 4. Discussion

Forward and reverse genetics approaches are two classical approaches often used to discover the gene functions underlying the phenotype. The classical forward genetic approach involves random mutagenesis and subsequent isolation of defective genetic mutants for a specific biological process or phenotype [5]. Whilst the frequency of mutations occurring spontaneously in nature is too low, physical and chemical mutagens are induced for accelerated plant breeding and are applicable to all plant species. Mutagens introduce random changes throughout the plant genome, resulting in manageable population sizes [29].

With the recent advances in next-generation sequencing technologies, high-throughput genome characterization has radically changed the value of TILLING. TILLING populations can be utilized either for visible developmental and morphological phenotypes or for scoring phenotypic behavior after the application of specific mutant screens [30]. The initial mutation detection method for TILLING was performed using a commercial denaturing HPLC method, referred to as Denaturing High-Performance Liquid Chromatography (DHPLC) [8], which was originally developed to examine a mutant population of barley [10]. The above method requires expensive equipment, but an enzymatic mismatch cleavage approach was successfully adapted for high-throughput mutant screening which employed the LI-COR system for the identification of Arabidopsis and wheat mutations [9,31,32,33,34]. Furthermore, a more economical and cost-effective substitute for the LI-COR system has been demonstrated with the use of conventional agarose or polyacrylamide electrophoresis [35,36]. In this study, we implemented a mutation scanning method based on the CEL I enzyme double-stranded cutting of mismatches with subsequent polyacrylamide electrophoresis, which reduced the outlay required to purchase the DHPLC system and the fluorescent primers as well as labor investment. Therefore, this method of mutation scanning technology will both increase the throughput and decrease the cost of rapid mutation discovery in mutagenized plants and other populations.

As described above, TILLING has the benefit of genome-wide mutation discovery and broader applicability for the pursuit of point mutations in any organism. Many TILLING populations have been constructed for numerous crop species using diverse mutagens. In barley, several TILLING populations were constructed and most of them were EMS-mutagenic [10,12,13,37,38]. Similar conclusions were drawn from other crop studies including rice, wheat, and maize [39,40,41]. Compared to other chemical and physical mutagenesis methods, EMS mutagenesis is more convenient and efficient since the mutation frequency of EMS appears to be high in all plant species and, based on historical accounts, appears to be independent of genome size [10,42]. Meanwhile, the mutant population development period is short, which makes it possible to generate a working reverse genetic population in approximately one and a half years, without having to carry out any plant transformation or selection of transformants [10]. In this study, 1022 mutant materials were obtained from the barley variety TX by EMS mutagenesis, and the morphological observation of the M_2_ and M_3_ generations showed that the mutant library was rich in morphological variation types, and the phenotypic mutation frequency reached 4.42%, involving leaves, plant height, stems, and spikes. This mutant library can provide ideal experimental materials for the study of functional genomes and breeding of barley.

We have previously shown that ion homeostasis plays a crucial role in resistance to waterlogging stress [43,44,45]. Thus, we were interested in identifying newly induced ion transporter mutants for exploration in our research program. In this study, we screened 5 gene mutants from this population and a total of 11 mutant strains were found. Compared to other gene mutants, the mutant number found in our study was low. In the study by Caldwell and his colleagues [10], there were six induced alleles of the *HvFor1* gene and four induced alleles of the *Hin-a* gene which were confirmed in the mutant pool. For the *Hvhox1* mutant, 31 alleles were identified, and the majority of these mutations (25 of 31) were G/C to A/T transitions [12]. Most of the analysis of induced mutagenesis focuses on characterizing non-synonymous changes or insertions/deletions. Interestingly, in our study, most of the mutant lines we found were synonymous. However, there are instances where synonymous base changes can induce a significant phenotypic effect, such as base changes within the miRNA binding site and possibly even synonymous changes, which could have dramatic effects on phenotypes.

TILLING of the *HvGLR3.5* gene in our newly developed TX cultivar TILLING population demonstrated the feasibility of generating an allelic series of a gene in barley with subtle phenotypic variation. In our study, three synonymous mutations were identified. Two of these mutations showed tuft and yellow striped leaves compared to the wild type. In *Marchantia polymorpha* L., *Mpglr-7* and *Mpglr-9* mutants were indistinguishable from the parental lines when grown in vitro or in soil, indicating that MpGLR activity is not required for normal vegetative growth and development [46]. In *Arabidopsis*, engineering the GLR3.3 channel by CRISPR impairs Ca^2+^/CaM-mediated desensitization which enhance plant systemic wound responses and anti-herbivore defense [47]. Meanwhile, GLR mutations in Arabidopsis produced tip growth defects in pollen tubes [48]. However, mutant phenotypes mentioned in *M. polymorpha* and *Arabidopsis* are typically the result of a single mutation; in the EMS mutant population, the mutant lines we explored carry multiple additional mutations in addition to those selected. Therefore, we observed various mutant phenotypes in the M_3_ generation between different allele mutants.

## 5. Conclusions

EMS mutagenesis can offer useful novel insights into the gene function in a species under investigation, and the availability of TILLING resources in different germplasms can be important. We developed a TILLING population in the two-rowed Chinese landrace TX9425 based on the CEL I enzyme with subsequent polyacrylamide electrophoresis, which decreased the cost and labor investment of rapid mutation discovery in mutagenized plants. In our population, the mutant phenotype variation was abundant, and we have demonstrated the availability for reverse genetics by identifying five ion transport-related gene mutations. This population will be a resource for functional analysis of genes and for high-throughput gene discovery in large-genome cereals.

## Figures and Tables

**Figure 1 plants-13-02490-f001:**
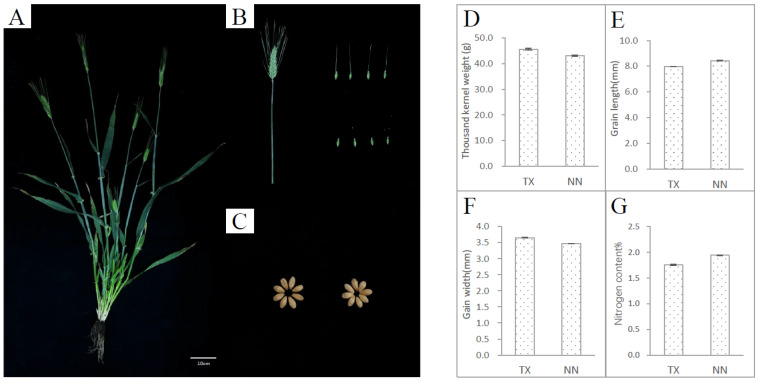
The phenotype of TX9425 (TX). TX is a Chinese two-rowed and hulled barley variety, which is mainly used for feeding. The image of the left panel (**A**) shows the whole plant after the filling period, and the top right image (**B**) shows the whole spike and seeds with awns. The bottom right image (**C**) shows the mature seeds. The thousand-kernel weight (**D**), relative water content (**E**), nitrogen content (**F**), and crude protein content (**G**) of TX and NasoNijo. NasoNijo (NN) is a Japanese two-rowed and hulled barley variety, which is mainly used for malting.

**Figure 2 plants-13-02490-f002:**
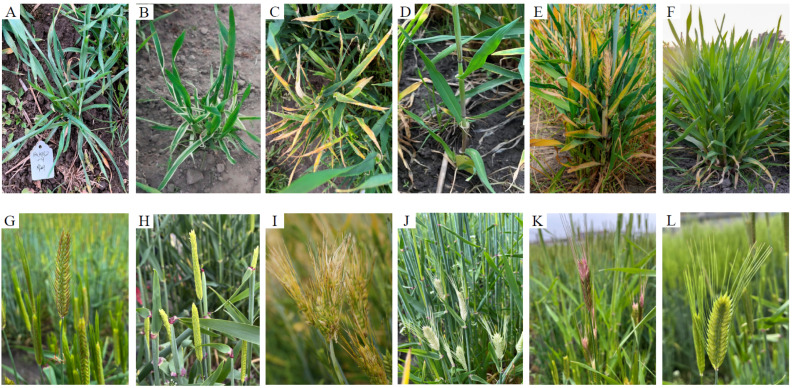
The phenotypes of M_3_ mutants in the field: (**A**) thin leaf, (**B**) leaf yellow stripes, (**C**) leaf tip yellow, (**D**) non-tillering, (**E**) multi-noded, (**F**) wild type of whole plant, (**G**) short awn, (**H**) awnless, (**I**) spike anomaly, (**J**) white glume, (**K**) red glume, (**L**) wild type of spike.

**Figure 3 plants-13-02490-f003:**
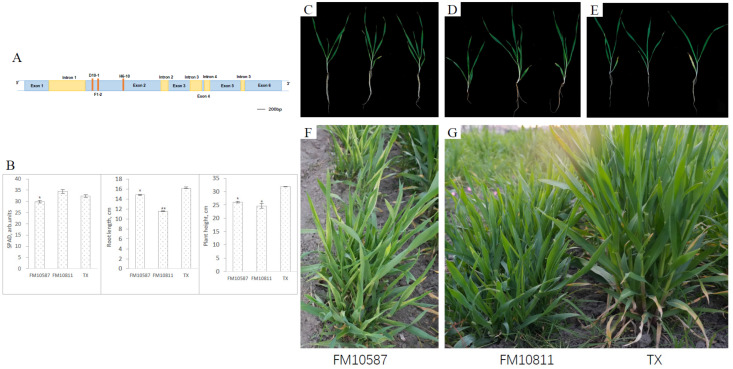
*Hvglr3.5* mutants detected from the TILLING population. The mutation locations (red main string) were shown in *HvGLR3.5* gene structure and all the mutations were synonymously (**A**). Two homozygous M_3_ plants, FM10587 and FM10811, were detected from the TILLING population. The chlorophyll relative content (SPAD), root length, and plant height of two mutants and TX were measured at the seedling stage (**B**). Single seedling in the pot ((**C**): FM10587; (**D**): FM10811; (**E**): TX) and seedling images in the field are shown ((**F**): FM10587; (**G**) left: FM10811; (**G**) right: TX). * indicates significant difference (*p* < 0.05), ** indicates significant difference (*p* < 0.01).

**Table 1 plants-13-02490-t001:** Mutant types and mutation frequency observed in M_2_ population.

Categories	Phenotype	Number of Mutants	Mutation Frequency
Seedling stage	Albino	2	0.03%
	Etiolated	12	0.17%
	Seedling tuft	34	0.47%
Leaves	Yellow striped leaves	36	0.50%
	Light leaf color	17	0.23%
	Yellow	41	0.57%
	White striped leaves	4	0.06%
	Flag leaf deformation	5	0.07%
	Leaf spot	16	0.22%
	Wide	4	0.06%
	Thin	9	0.12%
	Curled	4	0.06%
	Non-midrib	5	0.07%
Stems	Non-waxy	11	0.15%
	Non-tiller	11	0.15%
	Multiple tillers	15	0.21%
	Less tiller	23	0.32%
	Dwarf	9	0.12%
	Stem curvature	6	0.08%
Spikes	Six rows	2	0.03%
	Spike anomaly	16	0.22%
	White glume	13	0.18%
	Red glume	4	0.06%
	Long awn	3	0.04%
	Short awn	6	0.08%
	Awnless	5	0.07%
	Sterile	6	0.08%

**Table 2 plants-13-02490-t002:** Agronomic traits of some homozygous mutant lines in M_3_ generation.

Item	Plant Height (cm)	Spike Length (cm)	Spacer Number	Spike Number	Thousand-Kernel Weight (g)
TX9425	90.50	5.90	4.00	10.08	45.61
Mean of mutant lines	56.66	5.31	4.00	8.59	30.60
Maximum of mutant lines	91.50	12.03	5.00	15.56	48.39
Minimum of mutant lines	27.72	2.50	3.00	2.78	15.63
CV/%	13.29	16.38	0.11	43.04	12.08

**Table 3 plants-13-02490-t003:** List of TILLING targets and the number and distribution of mutation types.

Gene	Description	Mutation			
		Total	Intron	Synonymous	Non-Synonymous
*HvRBOHB*	Respiratory Burst Oxidase Homolog B	1	-	1	-
*HvRBOHE*	Respiratory Burst Oxidase Homolog E	2	-	1	1
*HvDTX16*	protein DETOXIFICATION 16-like	2	1	-	1
*HvGLR2.8*	Glutamate Receptor 2.8-like	3	-	1	2
*HvGLR3.5*	Glutamate Receptor 3.5-like	3	-	3	-

## Data Availability

Data are contained within the article and Appendix A.

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
