# Peer review of "Development and Characterization of a New TILLING Population for Forward and Reverse Genetics in Barley (Hordeum vulgare L.)"

_plants, 2024, doi:10.3390/plants13172490_

Round 1

Reviewer 1 Report

Comments and Suggestions for Authors

Line 35 delete either

Line 37 and 38, this sentence is not clear, needs to be revised

Line 79 change to landraces

Line 99 delete mainly, more details about traits need to be described here. Traits tested in M2 and agronomic traits of M3 need to be described in Maters and Methods.

Line 106 reference style here OK?

Line 129 wild type of TX, is TX a wild barley? not clear in the manuscript

Line 161 change to phenotypes

Line 162 29 traits, no description in Materials and Methods

Line 163 change to variations

Line 178 change to which is similar

Figure 3 ADE no legend

Line 236 change to reduction, seedling stage and in the field very confusing here, seedling stage can be also detected in the field. Is that meaning seedling stage in the glasshouse and adult plant stage in the field? Or other meanings?

Line 277 delete which

Comments on the Quality of English Language

English language is competent.

Author Response

Reviewer 1

Comments 1: Line 35 delete either.

Response 1: Thank you for pointing this out. “either” has been deleted.

Comments 2: Line 37 and 38, this sentence is not clear, needs to be revised.

Response 2: Thank you for pointing this out. This sentence has been changed to “With such a great diffusion as well as abundant genetic stocks that over 485000 germplasm accessions [3], the barley gene pool contain characters for wide environmental adaptability and strong stress resistance [4]”.

Comments 3: Line 79 change to landraces

Response 3: Thank you for pointing this out. “landraces“ has been changed.

Comments 4: Line 99 delete mainly, more details about traits need to be described here. Traits tested in M2 and agronomic traits of M3 need to be described in Maters and Methods.

Response 4: Thank you for pointing this out. The descriptions in M&M are on line 101-110.

Comments 5: Line 106 reference style here OK?

Response 5: Thank you for pointing this out. The reference style has been changed.

Comments 6: Line 129 wild type of TX, is TX a wild barley? not clear in the manuscript

Response 6: Thank you for pointing this out. TX is not a wild type, but a landrace. Usually, in mutant population ,we call the parent cultivar “wild type” which was used to compare with the mutants and this term was used in many other papers.

Feng X, Rahman MM, Hu Q, Wang B, Karim H, Guzmán C, Harwood W, Xu Q, Zhang Y, Tang H, Jiang Y, Qi P, Deng M, Ma J, Lan J, Wang J, Chen G, Lan X, Wei Y, Zheng Y and Jiang Q (2022) HvGBSSI mutation at the splicing receptor site affected RNA splicing and decreased amylose content in barley. Front. Plant Sci. 13:1003333. doi: 10.3389/fpls.2022.1003333

Jiang, C.; Lei, M.; Guo, Y.; Gao, G.; Shi, L.; Jin, Y.; Cai, Y.; Himmelbach, A.; Zhou, S.; He, Q.; et al. A reference-guided TILLING by amplicon-sequencing platform supports forward and reverse genetics in barley. Plant Communications 2022, 3, 100317, doi:10.1016/j.xplc.2022.100317.

Comments 7: Line 161 change to phenotypes

Response 7: Thank you for pointing this out. “phenotypes “ has been changed.

Comments 8: Line 162 29 traits, no description in Materials and Methods

Response 8: Thank you for pointing this out. The descriptions in M&M are on line 101-105.

Comments 9: Line 163 change to variations

Response 9: Thank you for pointing this out. “variations“ has been changed.

Comments 10: Line 178 change to which is similar

Response 10: Thank you for pointing this out. “which is similar” has been changed.

Comments 11: Figure 3 ADE no legend

Response 11: Thank you for pointing this out. The legend of Fig 3 is below the Fig 3 image.

Comments 12: Line 236 change to reduction, seedling stage and in the field very confusing here, seedling stage can be also detected in the field. Is that meaning seedling stage in the glasshouse and adult plant stage in the field? Or other meanings?

Response 12: Thank you for pointing this out. The TX and two mutants were both grown in the glasshouse and in the field. In the glasshouse, we measured the SPAD, root length and plant height physiological parameter at the seedling stage. In the field, we mainly observed the phenotype and harvested the seed.

Comments 13: Line 277 delete which

Response 13: Thank you for pointing this out. “which” has been deleted.

Reviewer 2 Report

Comments and Suggestions for Authors

As a traditional means in creating mutagenesis, using chemical mutagen can create lines with potential benefit mutations. Despite enormous work need to be done for screening the populations, the biggest advantage is that lines of chemical mutagenesis does not constrain by the regulation of GMOs. The present study creative a barley TILLING population contains 2000 lines. By screening mutant lines from M2&M3 generations, some phenotypes with distinct morphological traits have been identified. The created population is invaluable to breeding program. Also, the screening approach used in this study is cost-friendly and time-efficiently.

The manuscript can be accepted after the following issues are addressed:

1) L34: the word 'Hordeum' should be italic

2) section 2.1: If the TILLING population is created by using TX9425, what's the purpose to use variety NasoNijo as a material? Please explain it in this section.

3) Fig3: please add line names to panel C-E, or specify in the legend.

4) format of references list:

(i) genus name should be italic, eg. L344, L347

(ii) The title capitalization should be consistent, e.g. L345, L347, L351, L375, L384, L415-416, L433, L445, L454

(iii) some references have missing DOI.

Author Response

Reviewer 2

Comments 1:L34: the word 'Hordeum' should be italic

Response 1: Thank you for pointing this out. “Hordeum” has been changed.

Comments 2: If the TILLING population is created by using TX9425, what's the purpose to use variety NasoNijo as a material? Please explain it in this section.

Response 2: In Fig1, we want to describe the agronomic character of TX9425, and NasoNijo was used as control. NasoNijo is a two-rowed malting barley with good agronomic traits. If only showing the agronomic character of TX9425 without control, we can't learn the characteristics of this cultivar.

Comments 3:Fig3: please add line names to panel C-E, or specify in the legend.

Response 3: Thank you for pointing this out. The legend is added under Fig 3.

Comments 4: format of references list:

Response 4: Thank you for pointing this out. The references list is formatted.

Comments 5 :genus name should be italic, eg. L344, L347

Response 5: Thank you for pointing this out. The genus name was italic.

Comments 6: The title capitalization should be consistent, e.g. L345, L347, L351, L375, L384, L415-416, L433, L445, L454

Response 6: Thank you for pointing this out. The title you mentioned has been changed.

Comments 7:some references have missing DOI.

Response 7: Thank you for pointing this out. The missing DOI has been added.

Reviewer 3 Report

Comments and Suggestions for Authors

Dear the editor,

The submitted manuscript reports the new mutant population of Chinese landrace TX9425 in barley. In addition, this work reports the CEL1-based SNPs detection method on the mutant population. Although the barley mutant populations have been established in other works, making the tilling population in the genetic background suitable for both basic and applied agronomic researches is a worthwhile work.

Overall, the manuscript seems well documented, but I think some points below need to be reconsidered.

The first point is the analyses of HvGLR3.5 gene. The authors analyzed two lines, FM10587 and FM 10811. Both lines were shown to have a synonymous mutation, respectively.  The phenotypes of each line might be misunderstood to be caused by each of the mutations in HvGLR3.5 gene. As the authors discussed in the Discussion section, each of the EMS mutant lines have a lot of nucleotide substitutions. Thus, the phenotypes described in the manuscript in FM10587 and FM 10811 lines can be explained by mutations in other loci. In other words, if the authors are willing to explain the cause of the phenotypes in FM10587 and FM 10811 lines, additional experiments that exclude the possibility of the effects of other genetic loci must be conducted.

The second point is why the authors did not analyze the non-synonymous mutations. Since the authors successively found the non-synonymous mutations in genes interest, the phenotypic effects by the mutation may be interesting for the readers.

Minor points

1.     In abstract and other parts of the manuscript, “tuft” is used for describing a phenotype. However, it is difficult for me to image what is “tuft” phenotype.

Does it mean dwarf?

2.     In abstract, “with a genome smaller than those of other members of the Triticeae tribe”. Is this sentence correct? I think barley genome is smaller than those of the Triticeae crops. I am not sure about all of the species in the Triticeae tribe. The tribe includes more than 300 species.

3.     Line 210, provincial ?

4.     Fig S1. Arrows indicating 371bp and 956bp should be reversed.

Author Response

Reviewer 3

Comments 1:The first point is the analyses of HvGLR3.5 gene. The authors analyzed two lines, FM10587 and FM 10811. Both lines were shown to have a synonymous mutation, respectively.  The phenotypes of each line might be misunderstood to be caused by each of the mutations in HvGLR3.5 gene. As the authors discussed in the Discussion section, each of the EMS mutant lines have a lot of nucleotide substitutions. Thus, the phenotypes described in the manuscript in FM10587 and FM 10811 lines can be explained by mutations in other loci. In other words, if the authors are willing to explain the cause of the phenotypes in FM10587 and FM 10811 lines, additional experiments that exclude the possibility of the effects of other genetic loci must be conducted.

Response 1: Thank you for pointing this out. We agree with this comment. In EMS mutant, the phenotypes we observed may be not caused by the gene we tested which was also discussed in our manuscript. About the phenotype confirmatory experiment the reviewer mentioned, we are processing this experiment and the results will be published in another paper.

Comments 2:The second point is why the authors did not analyze the non-synonymous mutations. Since the authors successively found the non-synonymous mutations in genes interest, the phenotypic effects by the mutation may be interesting for the readers.

 Response 2: Thank you for pointing this out. We do want to analyze the non-synonymous mutations. Unfortunately, we haven’t got the pure mutants. The hvglr3.5 mutants we mentioned in the manuscript are pure and the phenotypes are stable.

Comments 3:In abstract and other parts of the manuscript, “tuft” is used for describing a phenotype. However, it is difficult for me to image what is “tuft” phenotype. Does it mean dwarf?

Response 3: Thank you for pointing this out. The word tuft means bush. The seedling is dwarf and like grass.

Comments 4: In abstract, “with a genome smaller than those of other members of the Triticeae tribe”. Is this sentence correct? I think barley genome is smaller than those of the Triticeae crops. I am not sure about all of the species in the Triticeae tribe. The tribe includes more than 300 species.

Response 4: Thank you for pointing this out. The “Triticeae tribe” is changed to “Triticeae  crops”.

Comments 5:   Line 210, provincial ?

Response 5: Thank you for pointing this out. This word has been deleted.

Comments 6:  Fig S1. Arrows indicating 371bp and 956bp should be reversed.

Response 6: Thank you for pointing this out. The mistake has been changed.

Round 2

Reviewer 3 Report

Comments and Suggestions for Authors

Dear the editor,

The authors have responded to all comments I made. I do not have any additional comments on the revised manuscript.